# Effect of Intake of Extra Virgin Olive Oil on Mortality in a South Italian Cohort with and without NAFLD

**DOI:** 10.3390/nu15214593

**Published:** 2023-10-29

**Authors:** Caterina Bonfiglio, Francesco Cuccaro, Angelo Campanella, Natalia Rosso, Rossella Tatoli, Gianluigi Giannelli, Rossella Donghia

**Affiliations:** 1National Institute of Gastroenterology—IRCCS “Saverio de Bellis”, 70013 Castellana Grotte, Italy; angelo.campanella@irccsdebellis.it (A.C.); rossella.tatoli@irccsdebellis.it (R.T.); gianluigi.giannelli@irccsdebellis.it (G.G.); 2Local Health Unit—Barletta-Andria-Trani, 76121 Barletta, Italy; francescocuccaroepi@gmail.com; 3Fondazione Italiana Fegato, AREA Sciegce Park, Basovizza, 34149 Trieste, Italy; natalia.rosso@fegato.it

**Keywords:** extra-virgin olive oil, non-alcoholic fatty liver disease (NAFLD), survival

## Abstract

Background: Extra-virgin olive oil (EVOO) is the main source of seasoning fat in the Mediterranean diet and it is one of the components with known protective factors on chronic-degenerative disease. We aimed to evaluate the effect of a medium-high level of oil consumption on mortality in a cohort with good adherence to the Mediterranean diet. Methods: A total of 2754 subjects who had completed the food questionnaire in the Multicenter Italian study on Cholelithiasis (MICOL) cohort were included in the study. EVOO consumption was categorized in four levels (<20 g/die, 21–30 g/die, 31–40 g/die, >40 g/die). We performed a flexible parametric survival model to assess mortality by EVOO consumption level adjusted for some covariates. We also performed the analysis on subjects with and without non-alcoholic fatty liver disease (NAFLD) to evaluate the effects of oil in this more fragile sub-cohort. Results: We found a statistically significant negative effect on mortality for the whole sample when EVOO consumption was used, both as a continuous variable and when categorized. The protective effect was stronger in the sub-cohort with NAFLD, especially for the highest levels of EVOO consumption (HR = 0.58 with *p* < 0.05). Conclusions: Our study has shown a protective effect of EVOO consumption towards all causes of mortality. Despite the higher caloric intake, the protective power is greater for a consumption >40 g/day in both the overall cohort and the sub-cohorts with and without NAFLD.

## 1. Introduction

The Mediterranean diet (MedDiet) is historically the traditional dietary pattern followed by the inhabitants of the Mediterranean region. It is characterized by high intakes of vegetables, fresh fruit, whole grains, legumes, nuts, fish, and extra-virgin olive oil (EVOO), while white meat and other animal foods such as dairy products and eggs are present in moderate amounts and red and processed meat are limited. Alcohol consumption in this diet is moderate and occurs mainly as red wine [1,2,3,4].

Defined for the first time by Ancel Keys in the 1960s [5], MedDiet is one of the most investigated dietary patterns today [6] for its potential protective and preventive role against chronic and degenerative diseases, cardiovascular diseases, metabolic syndrome, cognitive decline, and cancer [7]. A large Spanish multicenter randomized trial evaluated the effect of MedDiet in the primary prevention of CVD [8].

Other epidemiological studies showed an association between greater adherence to MedDiet and a reduction in incidence and mortality for major groups of chronic diseases and all causes of mortality [1,9,10]. An improvement in the quality of diet assessed by different indices, such as the Alternate Mediterranean Diet score, was associated with reduction in overall and CVD mortality over a period of 12 years [11].

A dietary pattern based on the model of MedDiet is recommended in patients with non-alcoholic fatty liver disease (NAFLD), an emerging public health issue. NAFLD is not only a major cause of liver-related morbidity and mortality worldwide, but also an independent risk factor for the development of noncommunicable diseases. No specific treatment has been defined for NAFLD, and the international guidelines suggest an approach based on lifestyle changes, such as following the Mediterranean dietary pattern [12].

The ancient inhabitants of Mediterranean countries have always cited olive oil as being responsible for the longevity of the population [6]. EVOO represents the main seasoning fat of this dietary model. It is the major source of unsaturated fatty acids, in particular oleic acid and linoleic acid. Other components are widely represented, including secoiridoids (oleuropein, oleacein, and oleocanthal), simple phenols (tyrosol and hydroxytyrosol), lignans (pinoresinol), flavonoids (apigenin), hydrocarbons (squalene), triterpenes (maslinic acid), and phytosterols (β-sitosterol) [13,14].

The polyphenols present in EVOO possess anti-inflammatory, antioxidant, neuroprotective, cardioprotective, anticancer, anti-obesity, antidiabetic, antimicrobial, and antisteatotic effects [8,14,15,16,17,18,19,20,21].

The composition and the phenolic concentration in EVOO depend on several factors, such as the specific type of tree cultivar, the climatic and growing conditions, the grade of maturity of the drupes at harvesting, and the extraction methods [22,23]. EVOO, unlike other olive oils, is always obtained by cold mechanical extraction and is characterized by a high concentration of antioxidant, anti-inflammatory, and anti-proliferative bioactive components and a low percentage of free fatty acids (<1%) [22].

The scientific literature is rich in studies that emphasize the protective effect of a moderate/high level of EVOO consumption on morbidity and mortality [1,2,3,4,14,15,16,17,18,19,20,21]. The primary aim of the present study is to investigate the effect of EVOO consumption on mortality in a specific cohort of a South Italian population, characterized by a high adherence to the Mediterranean diet and a medium-high consumption of EVOO. We also studied the effect of different classes of EVOO intake on the overall cohort and sub-cohorts with and without NAFLD.

## 2. Materials and Methods

Details about the study population have been published elsewhere [22,23] (Figure 1). In short, the analysis is based on the Multicenter Italian study on Cholelithiasis (MICOL), a prospective cohort study conducted by the Laboratory of Epidemiology and Biostatistics of the National Institute of Gastroenterology, “Saverio de Bellis” Research Hospital (Castellana Grotte, Bari, Italy). The MICOL Study is a systematic 1-in-5 random sample study drawn from the electoral list of Castellana Grotte (≥30 years old) in 1985 and followed up in 1992, 2005–06, and 2013–16. In 2005–06, using the same sampling scheme, this cohort was supplemented with a random sample of subjects (PANEL study) aged 30–50 years to compensate for the cohort ageing.

In this paper, the baseline for the MICOL cohort was established in 2005–06 to capture all ages and to homogenize follow-up time. All procedures performed were in accordance with the ethical standards of the institutional research committee [IRCCS Saverio de Bellis Research and Ethical Committee approval for the MICOL Study (DDG-CE-347/1984; DDG-CE-453/1991; DDG-CE-589/2004; DDG-CE 782/ 2013)] and with the 1964 Helsinki declaration.

### 2.1. Data Collection

Participants were interviewed at baseline in 2004–2005 by trained physicians and/or nutritionists to collect information on sociodemographic characteristics, health status, personal history, and lifestyle factors, including history of tobacco use, food intake, educational level (International Standard Classification of Education) [24], work profile (International Standard Classification of Occupations, International Labour Office) [25], and marital status. Weights and heights were measured in underclothing and without shoes. Weights were taken on an electronic balance, SECA©, and recorded to the nearest 0.1 kg. Height was measured with a wall-mounted stadiometer, SECA©, and recorded to the nearest 1 cm. Blood pressure (BP) measurement was performed following international guidelines [26]. The average of three BP measurements was calculated. The European Prospective Investigation into Cancer and Nutrition (EPIC) food frequency questionnaire (FFQ) was used to document the usual food intake of participants at baseline [27]. Nutritionists conducted an in-person structured interview asking participants to report on their frequency of usual intake of 233 foods items over the past year; they reported intakes per day, per week, or per year. They were also asked to estimate their portion sizes from photographs; questions were referred to the usual intake in the last year. No further dietary assessments were made during the cohort’s 12–13-year follow-up. Data from FFQs were entered by trained dietitians into the National Cancer Institute electronic database (Milan, Italy) to calculate the total intake of food groups, energy, and nutrients [28]. Scores for adherence to the MedDiet, separately and combined in the rMED synthetic score [28], were calculated using Stata statistical software version 16.0. A fasting venous blood sample was drawn, and the serum was separated into two different aliquots, the first of which was immediately stored at −80 °C, and the second was used to test the blood chemistry profile of the participants.

### 2.2. Exposure Assessment

The EVOO consumption was grouped into 4 categories: <20 g/die, 21–30 g/die, 30–40 g/die, and >40 g/die. We calculated a relative Mediterranean diet (rMED) score without EVOO to adjust for all the other components of the MedDiet in statistical analyses. The analyses were performed for sub-cohorts with and without non-alcoholic fatty liver disease (NAFLD) to assess a differential effect of EVOO. The outcome of this study is the all-cause mortality of the subjects enrolled in the MICOL cohort. Information about the vital status of participants was obtained from the Municipality of Castellana Grotte and was electronically linked with the database. Requests were also submitted to the municipalities of current residence for subjects who had migrated during the follow-up. Information about causes of death were obtained from the regional Nominative Registry of Causes of Death (ReNCaM).

### 2.3. Statistical Analyses

For analytical purposes, the EVOO consumption was grouped into <20 g/die, 21–30 g/die, 31–40 g/die, and >40 g/die.

Data are presented as mean (± SD), median (IQR) for continuous data, or frequency (%) for categorical data. Kolmogorov and Pearson’s chi-square tests were used to test differences between means and proportions.

Time from enrolment to death, moving elsewhere, or end of the study (31 December 2022), whichever occurred first, was considered as the observation time.

Since age is the most important risk factor for death, acting as a proxy for many unknown factors, we chose age at death as the time scale. We set 95 years of age as the maximum observation age to reduce comorbidity-related problems in the elderly in the more advanced age groups.

The Schoenfeld residuals were determined to assess the proportional hazard assumption.

A flexible parametric survival model, with three degrees of freedom, was fitted to the data to assess the association between the groups of EVOO consumption and all-cause mortality.

The same model was assessed for the group of subjects with and without NAFLD. This work used the EVOO consumption <20 g/die as a reference group.

Akaike’s criterion (AIC) and Schwarz’s criterion of Bayesian information (BIC) were run to include or not include each variable in the final model [29,30].

In the multiple adjustment model, we considered as potential confounders sex, KCAL without fats available, total cholesterol, HDL in range vs. not in range, AST, ALT, BMI, diastolic blood pressure, the relative Mediterranean diet score (rMED) with no oil consumption, and smoke habit. The Harrell C-index (C-Index) for censored data was employed to evaluate the predictive performance of any parameters helping to predict survival for all-causes mortality.

All statistical analyses were performed using Stata, statistical software version 18.0 (StataCorp, 4905 Lakeway Drive, College Station, TX 77845 USA); in particular, the stpmcr2 official Stata command was used, and its post-estimation commands allowed prediction of the sub-distribution hazard rate (SHR) for all-cause mortality.

## 3. Results

A total of 2970 subjects participated in the MICOL Study, of which 2754 (92.7%) had completed the food questionnaire. The study base generated a total observation time of 192,264.25 person-years.

In Appendix A the distribution of intake of the MedDiet nutrients by age at enrolment is shown. A downward trend in daily kcal, in relation to age, is observed, and the highest EVOO consumption is recorded in the 70–79 age group (44.43 (±19.14)). The increase in olive oil consumption is accompanied by an increase in overall caloric intake. A total of 675 (24.59%) subjects died during the observation time (16.9 years IQR (16.0; 17.3), 194 (28.7.1%) from cardiovascular diseases (CVD), 170 (25.2%) from neoplastic diseases, of which 17 (10.0%) were from malignant neoplasia of the colon, 18 were from malignant neoplasia of the liver (10.6%), 41 (6.07%) were from digestive system disease, and the remaining 270 (40.0%) were from other causes. Appendix A shows the consumption of food grouped into rMED categories. Calories decrease with the increase in rMED.

In Table 1, we show detailed information on the 2754 participants.

Table 2 shows detailed information about participants with and without NAFLD by EVOO consumption.

The results of the mortality hazard rates according to EVOO consumption categories for the whole sample and for the sub-cohorts of subjects with NAFLD are shown in Table 3.

There was a statistically significant negative effect on mortality for the whole sample when EVOO consumption was introduced as a continuous variable (HR 0.99, 95%CI 0.98; 0.99), and in analyses with categorized consumption, we observed two statistically significant effects: the first one for 31–40 EVOO intake (HR 0.73, 95% CI: 0.55; 0.97) and the second one for >40 g/die (HR 0.66, 95% CI: 0.50; 0.87).

Concerning the without-NAFLD sub-cohort, when EVOO consumption was introduced as a continuous variable in the model, there was no statistically significant effect on mortality, while in the NAFLD sub-cohort, HR was 0.99 (95% CI:0.98; 0.99).

In the without-NAFLD sub-cohort, we observed a statistically significant effect for EVOO consumption >40 g/die (HR 0.68, 95% CI: 0.48; 0.98). In the NAFLD sub-cohort, statistically significant effects were observed for 31–40 g/die (HR 0.58, 95% CI: 0.33; 0.999) and for >40 g/die (HR 0.58, 95% CI: 0.35; 0.97).

A downward trend was observed for the whole sample as well as for the two sub-cohorts.

Figure 2 shows the graphic representation using the post-estimation command of the mortality rates, in the form of HR of the two extreme EVOO consumption groups (<20 g/die and >40 g/die). Overall mortality is higher in the low consumption group compared to the high consumption group for all ages after 70 years and the gap grows with age.

Furthermore, the performances of the flexible survival model models were tested using the Harrell concordance, on the total cohort, patients with NAFLD, and patients without NALFD (C = 0.6319, 0.6096, and 0.6295 respectively).

## 4. Discussion

Our study confirms that extra-virgin olive oil is a food associated with longevity and MedDiet is an important pillar of preventive medicine. EVOO, as the major source of fat in this dietary pattern, is associated with benefits for human health, especially the cardiovascular system, against obesity, diabetes, and related metabolic disorders [12].

Many studies confirm the protective effect of moderate/high levels of EVOO consumption on morbidity and mortality. The disease group most closely investigated in terms of morbidity and mortality in relation to the consumption of olive oil is CVD, for which the protective relationship seems to be ascertained [31,32,33,34,35,36,37,38,39].

Other studies have taken into consideration mortality due to metabolic diseases such as type 2 diabetes, metabolic syndrome and metabolic-dysfunction-associated fatty liver disease, cognitive disorders, and tumors [39,40,41]. The existence of a clear protective effect against tumors is still controversial [40,41,42]. There are suggestions of a preventive effect against the onset of colorectal carcinomas, either through an anti-inflammatory and immunomodulatory action or through a modification of the intestinal microbiota [40].

In terms of biological mechanisms, the beneficial effects of EVOO on human health are attributed to antioxidant, lipid-lowering, anti-inflammatory, anti-aggregating, antiatherogenic, microbiota-improving, and insulin-modulating properties [43].

The MICOL cohort, which constitutes the population base of our study, is characterized overall by a good adherence to the Med Diet and by almost exclusive use of high-quality EVOO as a source of seasoning fat. The objectives of this study were to confirm the effects on reducing all-causes mortality with a medium-high consumption of EVOO and whether the protective effect is also confirmed for high intakes. To discriminate the effect of EVOO from that associated with the other components of MedDiet, an indicator of adherence to the Mediterranean diet (rMED) with the exclusion of EVOO was used.

In fact, high-quality EVOO contains many specific bioactive substances such as oleuropein, oleocanthal, tyrosol, hydroxytyrosol, and lignans, which show anti-inflammatory, antioxidant, and microbiota-regulating effects that are different from those contained in other healthy foods, with which they can interact synergistically [13,14]. We were also interested in exploring the effect of EVOO on mortality in the sub-cohorts with and without NAFLD, which is information present in the MICOL. EVOO is able to reduce hepatic fat accumulation independently from metabolic pathways, probably through increased fatty acid oxidation. Apart from the lower LDL oxidation, it is suspected that the main mechanism underlying olive oil beneficial effects in NAFLD includes a decrease in NF-κB activation and the improvement of insulin resistance (IR) [12].

Our results showed that in the MICOL cohort, EVOO has a protective effect both at moderate doses (31–40 g/die) and high doses (>40 g/die) compared to the lowest intake group (<20 g/die); this effect was observed in the whole cohort (0.73 95%CI 0.55–0.97 and 0.66 95%CI 0.50–0.87, respectively) and in the NAFLD sub-cohort (0.58 95%CI 0.33–0.99 and 0.58 95%CI 0.35–0.97, respectively).

Our results support the evidence that consumption of EVOO greater than 40 g/die produces a protective effect against many chronic-degenerative diseases, which has repercussions for a reduction in overall mortality.

This effect is evident despite the high-calorie intake due to larger quantities of fat-rich food such as oil. The importance of calorie restriction is known for its effects on longevity [44,45].

However, it has also been shown that MedD is the most recommended dietary pattern for NAFLD because it can reduce liver fat even without weight loss [46].

In balancing the advantages and disadvantages, we believe that a high consumption of EVOO oil can be suggested to the general population, and in particular to subjects suffering from NAFLD and more generally from hepatic steatosis (tested in the analyses, results not presented), since in the sub-cohort of subjects suffering from this condition the protective effect was even more evident.

The protective effect of high consumption compared to low consumption of EVOO is even more evident when the mortality rate by age is observed. These results showed that the protection is mainly seen in older adults, and the effect is even stronger in older adults and in subjects affected by NAFLD.

The strengths of our study are the cohort design and large population sample representative of an area with great adherence to the MedDiet. In fact, in this specific geographical location, the population is very “conservative” regarding eating habits, with constant behavior even in different age groups [47]. In addition, the exposure assessment was performed with a validated score [28] and a relatively long duration of follow-up.

However, this great adherence to MedDiet is also a limit because EVOO is practically the only seasoning fat used, with the exception of modest amounts of butter, lard, and margarine, so it is difficult to compare the effect of EVOO compared to other sources of fat.

However, a recent study has shown that replacing margarine, butter, mayonnaise, and dairy fat with olive oil was associated with a lower risk of mortality [48].

It should be noted that EVOO intake at baseline was high, and the assessment at baseline can be considered a good predictor of lifestyle habits in this cohort.

Finally, although the FFQ used has been validated, measurement errors are unavoidable, especially since it was impossible to assess the consumption of the different self-reported olive oil varieties. However, the Mediterranean population of southern Italy is characterized by the consumption of seasoning EVOO produced in the local area.

## 5. Conclusions

Our study showed a protective effect of EVOO consumption towards all causes of mortality. Despite the higher caloric intake, the protective power is greater for consumption >40 g/day in both the overall cohort and the sub-cohorts with and without NAFLD.

Our results therefore support the advice to take at least 40 g/day of high-quality EVOO within a varied and balanced diet, such as MedDiet, in the healthy population and in the population affected by NAFLD in the absence of health contraindications. However, this assumption may not be generalizable to population in other geographic areas where a MedDiet is atypical or which has different genetic and environmental characteristics.

Further studies are required to confirm the metabolic protective effects of EVOO and its components and to investigate specific biological pathways.

## Figures and Tables

**Figure 1 nutrients-15-04593-f001:**
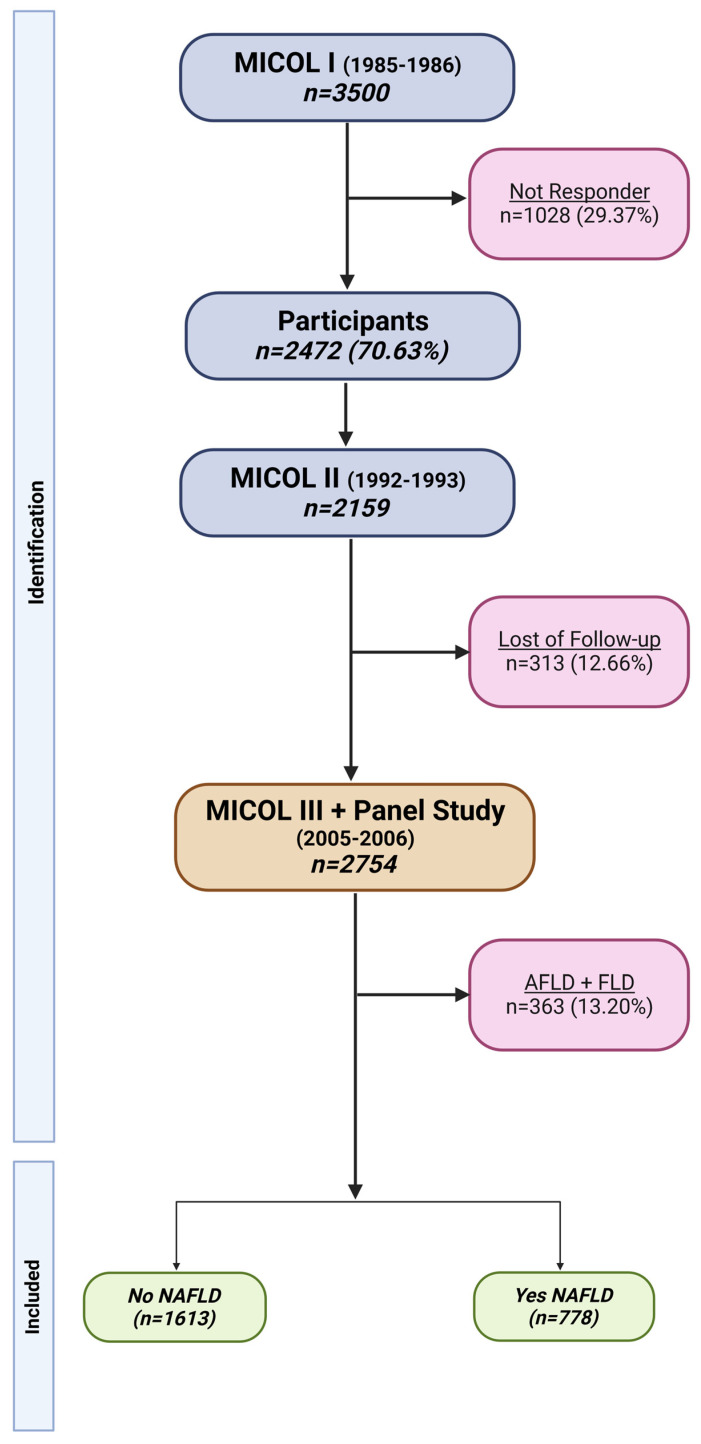
Flowchart of MICOL study and final cohort. This image was created with BioRender. (https://app.biorender.com/illustrations/65127f9a506f8015c32d0cb6, accessed on 26 September 2023).

**Figure 2 nutrients-15-04593-f002:**
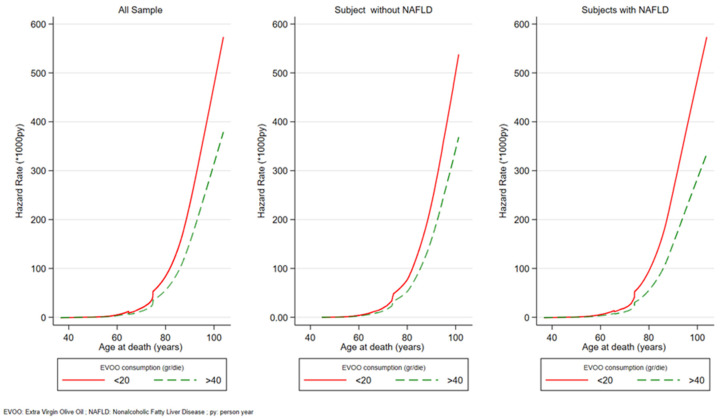
All-causes mortality distribution hazard rate (SHR) per 1000 py. MICOL Study. Castellana Grotte. (BA), Italy (2005–2022).

**Table 1 nutrients-15-04593-t001:** Characteristics of participants, by EVOO oil categories, of the MICOL/PANEL Study. Castellana Grotte. (BA), Italy (2005–2022).

N *** (2754)	EVOO Oil Categories (g/Die)				
	<20	21–30	31–40	>40	*p*–Value
	645 (23.42)	635 (23.06)	595 (21.60)	879 (31.92)	
Gender **					
Female	305 (25.6)	282 (23.6)	249 (20.9)	357 (29.9)	0.055
Male	340 (21.8)	353 (22.6)	346 (22.2)	522 (33.4)	
Age enrolment (yrs)	46.38 (13.00)	52.15 (13.99)	56.9 (14.77)	61.77 (13.54)	<0.001
Age categories (yrs)					
<40	256 (42.6)	153 (25.5)	103 (17.1)	89 (14.8)	<0.001
40–49	209 (37.0)	167 (29.6)	109 (19.3)	80 (14.2)	
50–59	84 (15.6)	130 (24.1)	145 (26.9)	181 (33.5)	
60–69	42 (8.3)	103 (20.4)	112 (22.2)	248 (49.1)	
70–79	37 (8.7)	65 (15.3)	87 (20.5)	235 (55.4)	
≥80	17 (14.3)	17 (14.3)	39 (32.8)	46 (38.7)	
DBP (mmHg) *	116.68 (18.67)	121.41 (18.56)	125.20 (19.55)	129.27 (19.86)	<0.001
SBP (mmHg) *	73.41 (10.23)	74.92 (10.77)	75.40 (9.89)	75.28 (9.69)	0.001
Weight (kg) *	74.58 (15.40)	74.55 (14.82)	75.53 (15.20)	75.75 (15.55)	0.31
BMI (kg/m^2^) *	27.71 (4.84)	28.34 (4.96)	29.00 (5.26)	29.55 (5.42)	<0.001
Kcal days *	2040.92 (712.00)	2161.72 (732.58)	2196.68 (692.40)	2348.09 (687.87)	<0.001
TGL (mmol/L) *	1.39 (1.05)	1.49 (1.20)	1.48 (1.03)	1.47 (0.91)	0.23
TC (mmol/L) *	5.17 (0.95)	5.15 (1.05)	5.19 (1.00)	5.14 (1.00)	0.78
HDL (mmol/L) *	1.33 (0.38)	1.32 (0.38)	1.30 (0.34)	1.33 (0.36)	0.36
LDL (mmol/L) *	3.22 (0.80)	3.13 (0.93)	3.21 (0.86)	3.13 (0.85)	0.064
Glucose (mmol/L) *	5.90 (1.40)	6.07 (1.48)	6.08 (1.58)	6.16 (1.51)	0.008
AST (μkat/L) *	0.20 (0.10)	0.21 (0.13)	0.22 (0.15)	0.23 (0.22)	0.001
ALT (μkat/L) *	0.29 (0.18)	0.30 (0.26)	0.30 (0.25)	0.30 (0.30)	0.95
Smoker ***					
Never/Former	498 (21.9)	516 (22.7)	502 (22.1)	756 (33.3)	<0.001
Current	147 (30.5)	119 (24.7)	93 (19.3)	123 (25.5)	
Age at Death (yrs) **	59.45 (53.66–68.98)	65.61 (56.95–77.96)	72.37 (60.34–81.60)	78.51 (70.02–85.05)	<0.001
Observation time ** (yrs)	16.91 (16.13–17.07)	16.93 (16.12–17.26)	16.94 (16.03–17.43)	16.97 (13.55–17.45)	0.033
Status ***					
Alive and/or Censored	562 (27.0)	516 (24.8)	436 (21.0)	565 (27.2)	<0.001
Dead	83 (12.3)	119 (17.6)	159 (23.6)	314 (46.5)	
Cause of death					
Colon Cancer	2 (11.8)	2 (11.8)	5 (29.4)	8 (47.1)	0.68
Liver Cancer	2 (11.1)	4 (22.2)	6 (33.3)	6 (33.3)	
OC	22 (16.3)	29 (21.5)	34 (25.2)	50 (37.0)	
CVD	20 (10.3)	29 (14.9)	43 (22.2)	102 (52.6)	
DSD	4 (9.8)	9 (22.0)	8 (19.5)	20 (48.8)	
OCD	33 (12.2)	46 (17.0)	63 (23.3)	128 (47.4)	
NAFLD					
No	402 (24.9)	371 (23.0)	335 (20.8)	505 (31.3)	0.67
Yes	188 (24.2)	182 (23.4)	177 (22.8)	231 (29.7)	
rMED	7.00 (6.00–9.00)	8.00 (6.00–9.00)	8.00 (7.00–9.00)	8.00 (7.00–10.00)	<0.001

AST: Aspartate Transaminase ALT: Alanine Amino transferase; BMI: Body Mass Index; TGL: Triglycerides; TC: Total Cholesterol; DBP: Diastolic Blood Pressure; SBP: Systolic Blood Pressure; HDL: High-Density Lipoprotein Cholesterol; LDL: Low-Density Lipoprotein Cholesterol; OC: Other Cancers; CVD: Cardiovascular Disease; DSD: Digestive System Disease; OCD: Other Causes Death. NAFLD: non-alcoholic fatty liver disease. rMED: relative Mediterranean Scoring System. Cells reporting subject characteristics contain * Mean ± (SD). ** Median (IQR). *** Number. (Percentage) Percentage calculated per row.

**Table 2 nutrients-15-04593-t002:** Characteristics of participants with and without NAFLD by EVOO oil categories. MICOL/PANEL Study. Castellana Grotte. (BA), Italy (2005-2022).

	with NAFLD				without NAFLD			
	EVOO Categories (g/Die)				EVOO Categories (g/Die)			
	<20	21–30	31–40	>40	<20	21–30	31–40	>40
N	188	182	177	231	402	371	335	505
Gender **								
Female	57 (21.4)	63 (23.7)	65 (24.4)	81 (30.5)	236 (28.8)	198 (24.1)	158 (19.3)	228 (27.8)
Male	131 (25.6)	119 (23.2)	112 (21.9)	150 (29.3)	166 (20.9)	173 (21.8)	177 (22.3)	277 (34.9)
Age enrolment (yrs)	46.80 (12.57)	52.60 (12.40)	55.79 (12.62)	61.57 (13.16)	45.50 (13.33)	51.31 (15.12)	55.47 (16.22)	61.21 (14.28)
DBP (mmHg) *	121.95 (18.18)	125.05 (18.16)	128.85 (18.50)	131.16 (19.09)	112.86 (18.10)	118.82 (19.01)	121.36 (18.91)	128.00 (20.32)
SBP (mmHg) *	76.85 (10.30)	77.41 (12.32)	78.15 (9.58)	76.89 (9.99)	71.08 (9.67)	73.22 (10.04)	73.40 (9.74)	74.55 (9.53)
Weight (kg) *	85.13 (14.68)	83.24 (14.46)	83.96 (16.52)	83.96 (16.04)	68.64 (12.76)	69.09 (12.34)	69.87 (12.50)	70.74 (13.36)
BMI (kg/m^2^) *	31.12 (4.60)	31.41 (4.82)	31.91 (5.49)	32.23 (5.68)	25.78 (3.99)	26.41 (3.90)	26.90 (4.29)	27.80 (4.68)
Kcal days *	1986.8 (723.6)	2185.4 (762.0)	2144.6 (726.1)	2299.0 (683.3)	2031.0 (680.3)	2117.7 (708.4)	2215.1 (680.5)	2340.4 (685.7)
TGL (mmol/L) *	1.87 (1.40)	1.88 (1.34)	1.90 (1.19)	1.85 (1.08)	1.11 (0.70)	1.17 (0.87)	1.16 (0.67)	1.26 (0.73)
TC (mmol/L) *	5.27 (0.96)	5.27 (1.16)	5.30 (1.02)	5.25 (0.93)	5.06 (0.89)	5.04 (0.97)	5.07 (0.97)	5.07 (0.98)
HDL (mmol/L) *	1.16 (0.28)	1.18 (0.29)	1.18 (0.28)	1.23 (0.27)	1.41 (0.38)	1.40 (0.40)	1.37 (0.36)	1.40 (0.38)
LDL (mmol/L) *	3.30 (0.82)	3.24 (1.09)	3.26 (0.81)	3.18 (0.83)	3.14 (0.76)	3.10 (0.86)	3.17 (0.86)	3.08 (0.83)
Glucose (mmol/L) *	6.26 (1.75)	6.22 (1.32)	6.34 (1.67)	6.52 (1.71)	5.62 (0.85)	5.85 (1.31)	5.80 (1.36)	5.89 (1.27)
AST (μkat/L) *	0.35 (0.18)	0.36 (0.25)	0.31 (0.14)	0.30 (0.18)	0.24 (0.13)	0.24 (0.25)	0.27 (0.25)	0.28 (0.34)
ALT (μkat/L) *	0.21 (0.13)	0.22 (0.10)	0.21 (0.08)	0.21 (0.09)	0.18 (0.07)	0.19 (0.10)	0.21 (0.16)	0.23 (0.25)
Status ***								
Alive and/or Censored	165 (27.5)	145 (24.2)	141 (23.5)	149 (24.8)	352 (28.6)	307 (24.9)	243 (19.7)	329 (26.7)
Dead	23 (12.9)	37 (20.8)	36 (20.2)	82 (46.1)	50 (13.1)	64 (16.8)	92 (24.1)	176 (46.1)
rMED	8 (6–9)	8 (6–9)	8 (7–10)	9 (7–10)	7 (6–9)	8 (6–10)	8 (7–10)	8 (8–10)

AST: Aspartate Transaminase ALT: Alanine Amino transferase; BMI: Body Mass Index; DBP: Diastolic Blood Pressure; SBP: Systolic Blood Pressure; TGL: Triglycerides; TC: Total Cholesterol; HDL: High-Density Lipoprotein Cholesterol; LDL: Low-Density Lipoprotein Cholesterol; rMED: relative Mediterranean Scoring System. Cells reporting subject characteristics contain * Mean ± (SD). ** Median (IQR). *** Number. (Percentage) Percentage calculated per row.

**Table 3 nutrients-15-04593-t003:** Hazard ratio (HR) and 95% confidence interval (95% CI) for all-causes mortality by EVOO g/die consumption.

	EVOO Consumption Categories (gr/die)							
	21–30		31–40		>40		Continuos	
	HR	95% CI	HR	95% CI	HR	95% CI	HR	CI 95%
All sample	0.82	0.61, 1.10	0.73 *	0.55; 0.97	0.66 *	0.50; 0.87	0.99 *	0.98; 0.99
Without NAFLD	0.76	0.51; 1.12	0.74	0.51; 1.07	0.68 *	0.48; 0.98	0.99	0.99; 1.00
With NAFLD	0.99	0.59; 1.69	0.58 *	0.33; 0.999	0.58 *	0.35; 0.97	0.99 *	0.98; 0.99

* *p*-value>0.05. <20 g/die referent categories. Adjusted for sex, KCAL without available fats, total Cholesterol, HDL in range vs. not in range, AST, BMI, DBP, rMED with no oil consumption and Smoke.

## Data Availability

The original contributions presented in the study are included in the article. Further inquiries can be directed to the corresponding author.

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
