# Peer review of "Effect of Intake of Extra Virgin Olive Oil on Mortality in a South Italian Cohort with and without NAFLD"

_nutrients, 2023, doi:10.3390/nu15214593_

Round 1

Reviewer 1 Report

Comments and Suggestions for Authors

This study concerned the effect of extra virgin olive oil (EVOO) intake on mortality in a South Italian cohort with and without non-alcoholic fatty liver disease (NAFLD). 2754 selected subjects (MICOL cohort) were inserted in the study, while four levels of EVOO consumption were categorized (ranging from <20g/die to >40 g/die). EVOO consumption (in particular at the level of >40g/die) was found to exert a negative effect on mortality for the whole sample, mostly in the sub-cohort with NAFLD.

Introduction deals with the protective and preventive role of the Mediterranean diet (MD), in which olive oil represents a fundamental component, for a series of diseases.

Materials and Methods refer to the prospective cohort MICOL study when considering the sampling of selected subjects for the present research. Data collection and exposure assessment are detailed, and statistical analyses are adequately illustrated.  

Results are efficaciously described by also including three tables and two figures.

Discussion points out on how high-quality olive oil is provided with important biological effects (e.g. antioxidant, anti-inflammatory and anti-atherogenic ones) through many bioactive substances. Interestingly, both moderate doses and high doses of EVOO (31-40 g/die and >40 g/die, respectively) resulted to be more protective in the two studied cohorts. Overall, the present research has a prevalent epidemiological significance, being further studies needed (as rightly observed by Authors) to better clarify the biological mechanisms by which the EVOO components exert their above favourable effects.

This study is somewhat interesting, having been performed with adequate accuracy and methodological programming. Some editorial adjustments should be adopted through the manuscript to make it more fluent and incisive. For example.  

-line 17:…for…=…four…

-lines 21-22:…when EVOO consumption was used both as a continuous variable, than was categorized…: this sentence is poorly comprehensive and needs an adequate reformulation.

-lines 138-139:…Time from enrollment…was the observation time…=… enrolment… moreover, please better explain.

-line 209:…on mortality, while in the NAFLD sub-cohort, HR…=…on mortality while, in the NAFLD cohort, HR…

Otherwise, lexicon, sentence conceptual adherence, “English style”, tables and figures with their legends, and references are adequate.

Comments on the Quality of English Language

Minor editing of English language required

Author Response

Requests attached

Reviewer 2 Report

Comments and Suggestions for Authors

The authors did a valuable study. Non-alcoholic fatty liver disease and its complications occupy one of the leading places in morbidity. We need simple and effective measures to prevent complications and treat NAFLD in addition to drug therapy. The advantages of the work are the wide coverage of participants both by age and pathology, as well as the stratification of groups by level of olive oil consumption. The work has great practical value and was carried out at a high scientific level. While reading the work, I had only a few minor comments.

1. Please, give the definition of the abbreviation MICOL both in the abstract (line 16) and in the subsection Materials and Methods (line 79), as well as PANEL (line 82) (if possible).

2. Also in the abstract and in keywords you need to decipher NAFLD (line 19)

3. Also decipher what rMed score means in the Materials and Methods subsection (line 122)

4. The text contains confusing abbreviations like OO (line 122) and EVO (line 175 and 187), decipher them or replace them with EVOO.

5. In Table 1, unify the decimal separator, is it period or comma?

6. In Table 2, it seems to me, there are two table headers.

7. There is no point in introducing the abbreviations T2D and MAFLD, because they are not used anywhere else (lines 238 and 239).

8. What does IR (line 264) mean?

Author Response

Requests attached

Reviewer 3 Report

Comments and Suggestions for Authors

Effect of intake of extra virgin olive oil on mortality in a South Italian cohort with and without NAFLD, examines a group of subjects that at the time of initial study were generally adherent to the Mediterranean diet, as defined by Ancel Keys and rated by rMED, for the impact of EVO consumption levels on all-cause mortality during a 17 year follow up.

The study findings suggest an effect of EVO on morality, particularly in the high consumption group given the apparent limitation that the subjects are adhering to a particular diet regime with some variation.

There are a few points that need clarification.

Methods

In Figure 1 and related text.  Are any subjects from MICOL I and MICOL II included in the MICOL panel studied in this paper?  It is not clear.

On line 110, intakes per day, per week, “or” per year were reported.  Is this really OR, or is should this be AND? If it is OR, how am I supposed to rate the methodology?  Also, was there any follow up for any MICOL cohort on changes in diet habits with age and what was the result?

On line 116, can you mention the fraction of the subjects that failed the rMED criteria at recruitment and some of the main diet factors that contributed to not meeting the rMED criteria as well as the cut-off number in the rMED scale that defines adherence or non-adherence?

Line 150-AIC and BIC are mentioned here. What impact did this analysis have on inclusion or exclusion of any parameters?  No mention of this in results.

My main concern is the possible age-related shift in diet habits that appears in the rMED scores across age groups and the EVO levels consumed across age groups.   These stand out Table 1 and caused me to speculate on meaning and possible causation that might impact interpretation of the study findings.  My main speculative interpretation is that there might be some degradation in diet practices with time.  That is the older groups were raised when the diet and use of EVO was “enforced” by a variety of factors, cultural, economic, educational and this is being lost in the younger subjects, particularly the use of EVO vs. dairy products and so adherence to the initial rMED rating might be lost in the younger sets in the follow up interval, where the older subjects have presumably adhered over these same points in their lives.  I have reviewed papers on the impact of college level education on alteration in diet-from a negative view- where the college environment encouraged a shift to the cafeteria diet with junk foods and dairy products, that may not be completely salvaged in the post-college interval. I was wondering if evidence for such a shift in EVO consumption and the rMED rating in the entire population shows higher rMED score and higher EVO consumption for MICOL I vs. MICOL II vs. MICOL III?  This may or may not impact your conclusions.  However, I do wonder whether within age groups, say the 50-79 years of age sets (high EVO consumption in general) and <40 to 49 years of age sets (lower EVO consumption) what are the findings for all cause mortality vs. EVO levels?  

Anyway, please comment in the text on these matters.

Minor-Fig. 3 could be dropped. The information is found in the text. 

Comments on the Quality of English Language

OK

Author Response

Requests attached

Round 2

Reviewer 1 Report

Comments and Suggestions for Authors

According to the suggestions on my previous review, Authors now appear to have satisfactorily emended the manuscript.

Reviewer 3 Report

Comments and Suggestions for Authors

The authors have addressed my concerns adequately.